# CX-5461 Treatment Leads to Cytosolic DNA-Mediated STING Activation in Ovarian Cancer

**DOI:** 10.3390/cancers13205056

**Published:** 2021-10-09

**Authors:** Robert Cornelison, Kuntal Biswas, Danielle C. Llaneza, Alexandra R. Harris, Nisha G. Sosale, Matthew J. Lazzara, Charles N. Landen

**Affiliations:** 1Department of Obstetrics and Gynecology, University of Virginia, Charlottesville, VA 22908, USA; jrc3hg@virginia.edu (R.C.); kb4zg@virginia.edu (K.B.); dcl5z@virginia.edu (D.C.L.); alexandra.harris2@nih.gov (A.R.H.); 2Department of Chemical Engineering, University of Virginia, Charlottesville, VA 22908, USA; ns5pr@virginia.edu (N.G.S.); mjl9cd@virginia.edu (M.J.L.)

**Keywords:** ovarian cancer, ribosomal synthesis, RNA polymerase I, CX-5461, chemoresistance, patient-derived xenograft, STING, cytosolic DNA

## Abstract

**Simple Summary:**

CX-5461 is an RNA polymerase I inhibitor that is in clinical trials for both advanced hematological cancers and solid tumors. Experimentally, this drug has been shown to induce a p53-independent DNA damage response through ATM and ATR kinase, and has particular activity against chemoresistant tumors. The current study shows for the first time that CX-5461 treatment in ovarian cancer cells induces the release of cytoplasmic DNA that stimulates cGAS–STING signaling, leading to the production of IFN type I in both cancer cells and xenografts in vivo. Because the cGAS–STING pathway is a key mediator of the immune response against cancer cells, this novel finding may lead to utilization of RNA Pol I inhibitors in combination with checkpoint inhibition.

**Abstract:**

Epithelial ovarian cancer (EOC) is the deadliest of the gynecologic malignancies, with an overall survival rate of <30%. Recent research has suggested that targeting RNA polymerase I (POL I) with small-molecule inhibitors may be a viable therapeutic approach to combating EOC, even when chemoresistance is present. CX-5461 is one of the most promising POL I inhibitors currently being investigated, and previous reports have shown that CX-5461 treatment induces DNA damage response (DDR) through ATM/ATR kinase. Investigation into downstream effects of CX-5461 led us to uncovering a previously unreported phenotype. Treatment with CX-5461 induces a rapid accumulation of cytosolic DNA. This accumulation leads to transcriptional upregulation of ‘STimulator of Interferon Genes’ (STING) in the same time frame, phosphorylation of IRF3, and activation of type I interferon response both in vitro and in vivo. This activation is mediated and dependent on cyclic GMP–AMP synthase (cGAS). Here, we show THAT CX-5461 leads to an accumulation of cytosolic dsDNA and thereby activates the cGAS–STING–TBK1–IRF3 innate immune pathway, which induces type I IFN. CX-5461 treatment-mediated immune activation may be a powerful mechanism of action to exploit, leading to novel drug combinations with a chance of increasing immunotherapy efficacy, possibly with some cancer specificity limiting deleterious toxicities.

## 1. Introduction

Ovarian cancer remains the deadliest gynecologic malignancy and 40 years of research has seen little change in overall survival rates, with the exception of PARP inhibition in BRCA-mutated tumors [1,2]. The standard treatment for advanced ovarian cancer is primary cytoreductive surgery followed by chemotherapy. Despite initial responsiveness to combination chemotherapy of carboplatin and paclitaxel, the occurrence of chemo-resistant tumors is a major barrier, and therefore novel therapeutic options capable of attacking chemoresistant cells are a major goal in this field. The primary reason for chemotherapy resistance is still unknown beyond the expected heterogeneity of cancer cells. Recently, both our group and others have demonstrated that cell populations resistant to cytotoxic chemotherapy have shown an increase in ribosome biogenesis, making RNA polymerase I a possible drug target for chemoresistant EOC [3,4].

Transcription of the ribosomal RNA genes (rDNA) is mediated by RNA polymerase I (Pol I) and is a key regulatory step for ribosomal biogenesis (RiBi). Increased ribosomal biogenesis has been shown to be a hallmark of tumor malignancy for decades [5]. One of the first pathologic hallmarks of cancer was in fact pronounced nucleoli, generally indicating a high degree of RiBi. Dysregulation of the key components of RiBi has been found in many cancers [6]. Therefore, targeting RNA polymerase I could be a potential therapeutic strategy, especially in chemoresistant tumors. There are currently two known inhibitors of RNA polymerase I, CX-5461 and BMH-21 [7,8]. CX-5461 is the first selective Pol I inhibitor that has finished phase I clinical trials, with promising results in advanced hematological cancer treatment [9]. CX-5461 has been shown to cause G-quadruplex stabilization, DNA damage, ATM/ATR-mediated DNA damage response activation, senescence, and autophagy [10].

In this study, we characterize a previously unreported consequence of CX-5461 treatment: we show that CX-5461 treatment induces stimulator of IFN genes (STING) signaling and its associated transcription programs in ovarian cancer cells. We show that CX-5461 induces a rapid and robust buildup of cytoplasmic DNA. The cytosolic DNA accumulation by CX-5461 leads to downstream activation of the cytosolic DNA detection system and response pathways, including STING. Cytosolic DNA is detected by the cyclic GMP–AMP synthase (cGAS) and activates STING signaling, which leads to a type I interferon response. CX-5461 induces transcription of CXCL10 and IL-6. Knockdown of STING and cGAS abolishes CX-5461-induced transcription of CXCL10 and IL-6. We further show that CX-5461 in a xenograft model decreased tumor burden, increased STING staining and activates transcription of certain cytokines. This is significant because therapies that activate STING, especially ones with some cancer selectivity, may be a powerful mechanism to exploit in order to increase the efficacy of immunotherapies that can lead to durable cures.

## 2. Materials and Methods

### 2.1. Reagents and Cell Culture

CX-5461(Pidnarulex) was purchased from Selleckchem (cat# S2684, Houston, TX, USA)—the structure and details are available in (https://www.selleckchem.com/products/cx-5461.html accessed on 1 October 2021)—and dissolved in 50 mmol/L NaH_2_PO_4_ (pH 4.5) to make a 5 mmol/L stock solution. HeyA8 and COV362 (ovarian cancer cell lines) were used in this study and all cells were maintained in RPMI-1640 medium supplemented with 10% FBS (Hyclone, Logan, UT, USA). All ovarian cancer cell lines were routinely screened for Mycoplasma species (ATCC; Universal Mycoplasma Detection Kit ATCC, 30-1012K), with experiments carried out at 70% to 80% confluent cultures. Purity of stock cell lines was confirmed with short tandem repeat genomic analysis, and all cell lines were used within 20 passages from stocks.

### 2.2. RNAseq

Illumina Truseq was performed on cell lines after RNeasy RNA extraction and purification (Qiagen, Dusseldorf, Germany). RNA integrity was verified using the Agilent TapeStation. After sequencing, raw data files were checked using FastQC (FastQC, RRID:SCR_014583, Version 0.11.9 [11]. Gene set association analysis was performed by importing counts from Kallisto (v0.46.1, 2019.), alignment to tximport for count table extraction and into GSAAseq [12].

### 2.3. siRNA Transfection

siRNA was transfected into different ovarian cancer cell line (HeyA8 and COV362) cells using RNAi- MAX (Invitrogen, Carlsbad, CA, USA) as per the manufacturer’s protocol. Briefly, 5 × 10^5^ cells were seeded in a 10 cm dish in growth medium. Next day, transfection complex (siRNA and RNAiMax) was added to the cells after the cells were placed in OptiMEM without serum, the cells were incubated for 4 to 6 h, the transfection complex was then removed and washed with PBS, and the growth medium was added. The siRNA sequences are in Appendix A.

### 2.4. Immunoblotting and Immunofluorescence

STING (Cell Signaling Technology Cat# 13647, RRID:AB_2732796), IRF3 (Cell Signaling Technology Cat# 11904, RRID:AB_2722521) and pIRF3(S396) (cat#29047) antibodies were purchased from Cell Signaling (Boston, MA, USA) and procedures for immunoblotting and immunofluorescence were performed as in our previous publication [3]. For R-loop immunofluorescence, the S9.6 antibody was purchased from Kerafast (Cat# ENH001, RRID:AB_2687463, Boston, MA, USA). The immunofluorescence protocol was modified, and the cells were fixed with 4% paraformaldehyde in PBS for 5 min and permeabilized with 0.01% saponin in PBS. Live staining with Sir-Hoechst (Spirochrome Inc, cat#SC007, Stein am Rhein, Switzerland) was performed following manufacturer instructions at 1 µM concentration in media. Picogreen staining was performed according to manufacturer instructions using Quant-iT^TM^ Picogreen^TM^ dsDNA Assay Kit (Cat#P7589. DAPI was purchased from Thermofisher Scientific (cat#62247, Waltham, MA, USA). For quantification, images were taken from 5 random positions across 2 slides of each treatment, positive cytoplasmic DNA staining was counted and representative images were shown. Three independent experiments were performed.

### 2.5. Xenografts

All mouse work was performed after protocol submission, review, and approval by the UVA Institutional Animal Care and Use Committee (IACUC, ACUC). Nude mice were implanted with 1 × 10^6^ tumor cells (COV362 or HeyA8), as per our previous publication [3], treated once a week for 3 weeks by oral gavage using 50 mg/kg CX-5461 in 50 mmol/L NaH_2_PO_4_ (pH 4.5) or by vehicle only. Tumors were collected, weighed, submerged in TPER (Thermo, cat# 78510, Waltham, MA, USA), and homogenized using an Omni tissue homogenizer (Omni International, El Cajon, CA, USA) for 3 bursts of 15 s. Protein was quantitated using BCA assay (Pierce, cat# 23225, Rockford, IL, USA). Luminex was performed per manufacturer instructions with normalization to protein concentration.

### 2.6. Immunohistochemistry

Formalin-fixed paraffin-embedded tumors were sectioned and stained with an anti-STING antibody (Cell Signaling, cat#13647) using the manufacturer’s recommendations. Completed slides were read by a staff pathologist.

### 2.7. Statistics

For testing statistical significance comparing each drug to control, we chose a one-way ANOVA (SDs were significantly different) with Dunnett’s test for multiple hypothesis testing correction. Luminex assays were performed per the manufacturer’s protocols, with each analyte having the coefficient of variation of standard curve replicates at each dilution level determined. Chi-Square was also determined for the distance between observed concentrations and expected concentrations.

## 3. Results

### 3.1. CX-5461 Induces Cytoplasmic DNA Accumulation

Previous work in our lab and by other investigators has demonstrated that CX-5461 treatment markedly induces DNA damage response, leading to G2 phase delay [3,13]. While further investigating the effects of CX-5461 in live cells, we noticed the accumulation of cytoplasmic DNA. COV362 serous ovarian cancer cells were treated with either vehicle or CX-5461 for 5 h, stained with Sir-Hoechst and imaged live. CX-5461 treatment showed dramatic buildup of cytosolic DNA staining (Figure 1A). To better understand the timing of cytoplasmic.

Other cytotoxic DNA-damaging agents have also been shown to induce the release of cytoplasmic DNA [14]. To examine the effect of other drugs used clinically against ovarian cancer, we treated COV362 cells with IC50 dosages of CX-5461, BMH-21, carboplatin, paclitaxel, or temozolomide and stained with Sir Hoechst. Cells with positive cytoplasmic DNA staining were counted and quantified (Figure 1E). CX-5461 treatment significantly induced the cytoplasmic DNA positive cells compare to other drugs (*p* < 0.001).

### 3.2. CX-5461-Mediated Cytosolic DNA Induces the cGAS/STING System

Cytosolic DNA from genotoxic damage and pathogen infection is detected by a complex system leading to a type I interferon response [15,16,17]. To examine transcriptomic evidence of activation of the STING system, RNA-seq analysis of HeyA8 cells treated with CX-5461 for 24 h was performed. Downstream analysis using gene set association analysis of sequence count data (GSAAseqSP [12]) showed the expected loss of ribosome signaling from CX-5461 treatment, an upregulation of the cytosolic DNA detection system, as well as activation of IFNA signaling (Figure 2A).

The cytosolic DNA sensor cyclic GMP–AMP synthase (cGAS) is one of the most powerful activators of the STING signaling pathway. After the recognition of cytosolic DNA, cGAS activates STING, and in turn induces phosphorylation and nuclear translocation of IFN transcriptional regulatory factors TANK-binding kinase 1 (TBK1) and IFN regulatory factor 3 (IRF3). To investigate the cytosolic DNA-mediated IFNA signaling, we first evaluated the status of STING in HeyA8 cells. HeyA8 cells were treated with 2 µM of CX-5461 over a time course starting 15 min after treatment and monitor STING activation by immunoblotting. CX-5461 treatment increased the level of STING in a time-dependent manner (Figure 2B). Activation had ceased by 24 h. STING activation leads to phosphorylation of IRF3, which in turn leads to type I interferon activation [18]. To check the downstream signaling, HeyA8 and COV362 cells were treated with CX-5461 and cells were collected at different points and immunoblotting performed for indicated proteins. Phosphorylation of IRF3 in both HeyA8 and COV362 cell lines was observed after 1 h of CX-5461 treatment (Figure 2C). IRF3 is a transcription factor, where upon phosphorylation, it translocates to the nucleus and activates the transcription of its target genes, such as CXCL10, CCL3 CCL5, as well as a variety of interleukins, such as IL-6, depending upon the cell types. Therefore, we then examined mRNA expression of CCL5 and CXCL10, two major target genes downstream of STING activation. We found a significant increase in IL-6 and CXCL10 mRNA levels after CX-5461 treatment (Figure 2D) in both HeyA8 and COV362 cells.

We speculated that CX-5461 induces accumulation of cytosolic dsDNA and stimulates chemokine expression through the activation of the cGAS–STING–TBK1–IRF3 signaling pathway. To explore that possibility, we depleted two major components cGAS and STING in HeyA8 and COV362 cells by siRNA-mediated knockdown (Figure 3A). Depletion of either cGAS or STING failed to induce CX-5461-mediated induction of CXCL10 and IL-6 (Figure 3B).

### 3.3. In Vivo CX-5461 Treatment Induces Type I Interferons

To examine the effect of CX-5461 in vivo, we subcutaneously (SQ) implanted immunocompromised mice with COV362 or HeyA8 cells. After tumors grew to a size of at least 0.5 cm in dimension, mice were treated with CX-5461(50 mg/kg) or vehicle (Na_2_PO_4_ pH-4.5) by oral gavage weekly. Tumors were serially measured and harvested after 3 weeks, and cytokine profiling was examined by Luminex. Overall tumor burden was decreased by CX-5461 monotherapy even with just three treatments (Figure 4A). Luminex innate immune profiling showed an increase in Eotaxin, CCL3, CXCL10, and IL-1b in COV362 SQ (Figure 4B). Consistent with our in vitro findings, HeyA8 cells showed a moderate cytoplasmic increase in STING staining after treatment (Figure 4C).

## 4. Discussion

Our preliminary data have shown that the POL I inhibitor CX-5461 induces a significant accumulation of cytosolic DNA, transcriptionally activates STING, and induces phosphorylation of IRF3, which induces type I IFN in ovarian cancer cells. Furthermore, CX-5461 treatment decreases tumor burden in a xenograft model. Our data uncovered that DNA damage induced by CX-5461 generates cytosolic DNA, primarily dsDNA. CX-5461 treatment induces secretion of the type I interferon-associated cytokines: IL-6 and CXCL10. Previously, PARP inhibitor, BMN673 has been shown to activate the induction of type I IFNs such as IL-6 and CXCL-10 through the DNA-sensing cGAS–STING pathway [19].

STimulator of Interferon Genes (STING) is a signaling molecule that plays a vital role in controlling the transcription of many host defense genes, including pro-inflammatory cytokines and chemokines, and type I interferons (IFNs) [20,21]. STING expression is usually inhibited or lost in many cancers; however, at the same time, it is also detected at different levels in some tumors [22]. STING protein in cancer cells permits examining the consequences of its activation by its agonists, such as cGAMP [23] and dsDNA [24]. Those studies revealed that the function of the pathway or the ability to activate downstream proteins is frequently defective in tumors cells. It has also been reported that the majority of ovarian cancer cells exhibit defective STING signaling [25]. However, we found that CX-5461 treatment induces the activation of STING signaling both in vitro and in vivo. This activation occurs in our system at the 60 min mark, coinciding with the presentation of the cytoplasmic DNA microvesicles. This activation appears to be short lived in terms of how long the initial activation lasts, and how long inflammatory cytokines are generated. This timing needs to be further elucidated as it could have profound effects on whether pro- or anti-tumor inflammation occurs [26]. While this short-lived response could be detrimental to checkpoint inhibitor synergy, it also suggests a highly tunable effect that could be modulated by small doses specifically used to induce and maintain type I inflammatory signaling in tumors where we see anti-tumor immune responses from STING activation.

The activation of STING has many clinical implications. One of the main consequences of STING activation is to mediate inflammatory and anti-viral cellular programs by engaging the transcription factor IRF3. Another major signaling element initiated by STING is NF-κB-mediated transcriptional activation. Moreover, STING activation also initiates events that support anti-proliferative cell states such as senescence, apoptosis and pyroptosis. Canonical autophagy is depend on the ULK complex and TBK1, and is involved in the control of STING-mediated autophagy. Recently during replicative stress, cells were found to engage the STING–autophagy pathway to induce the autophagic cell death program, thereby inhibiting tumor growth [27,28]. Further study is necessary to reveal whether CX-5461 also has a role to induce the STING–autophagy pathway, and thereby regulate the outcome.

CX-5461-mediated STING activation could be used to promote stronger apoptotic responses in TP53-mutated tumors through activation of cytokine-dependent cell death pathways such as IRF3-mediated mitotic cell death and NLRP3 inflammasome-dependent pyroptosis [29]. Specific cytokines activated through CX-5461 treatment could be used to enhance immune checkpoint inhibitor responses, a critical therapeutic option that remains elusive in HGSOC. CXCL10 has been shown to potentiate checkpoint inhibitors in HR-deficient tumors, a common presentation in HGSOC patients and one that is already determined prior to treatment with PARP inhibitors [30]. CXCL10 and interferons (IFNG) have also been shown to enhance lymphocyte infiltration in some tumors and HGSOC patients often have poor T-cell infiltration at presentation. With short-acting, specific dose scheduling of CX-5461, one could modulate the tumor microenvironment of poorly infiltrated tumors to a microenvironment that is more anti-tumor, in terms of immune and inflammatory response [31,32]. Much more work will need to be done to fully understand the role that this specific type of STING activation plays in modulating the tumor immune environment and its responses to checkpoint inhibition.

There are critical questions remain requiring further study. HeyA8 cells are wild type for TP53, while COV362 cells have the TP53 (Y220C) mutation. This could explain the differences in response, since TP53 has been implicated in regulating immune responses, as well as response to cytosolic DNA specifically. Further, work needs to be done to elucidate the role of TP53 in response to CX-5461 and how the different mutations change the outcome. The inflammatory phenotype may be a STING-dependent activation, but it could also be partly associated with a senescence-associated secretory phenotype (SASP) as CX-5461 induces senescence in many cell types [28,29]. Most importantly, does the cancer specificity attributed to CX-5461 apply to this STING activation? As we saw variability in response in different cell lines, a comprehensive analysis of multiple cell subtype and a biomarker for response are needed. A cancer-specific STING activator would be a monumental discovery in terms of increasing the number of patients responding to immunotherapy and moving immunotherapies from temporary efficacy to durable cures.

## 5. Conclusions

We have shown that CX-5461 induces the release of cytosolic DNA, which is recognized by cGAS, and then cascade signaling occurs (Figure 5). Although the importance of the cGAS–STING pathway in dictating anti-tumor immunity through potentiating type I IFN production was recently shown, this study may provide additional insight into CX-5461-mediated anti-tumor activity through the cGAS–STING pathway which can be implicated in cancer therapy.

## Figures and Tables

**Figure 1 cancers-13-05056-f001:**
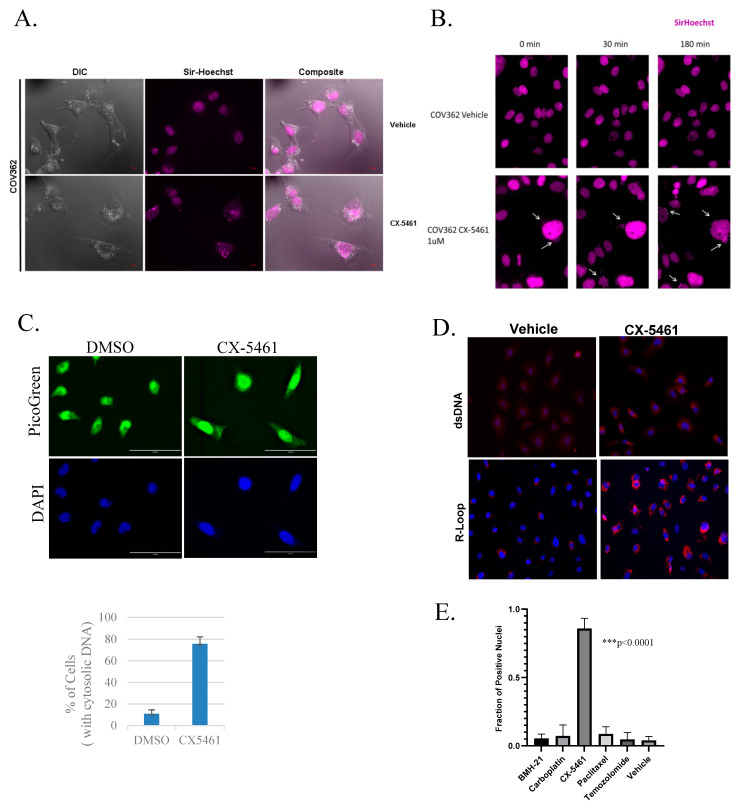
CX-5461 induces cytoplasmic DNA accumulation. (**A**) COV362 cells treated with 1 µM CX-5461 or vehicle for 5 h, stained with Sir-Hoechst and imaged live with a 63× objective. (**B**) Sir-Hoechst pre-labeled nuclei were treated with vehicle or 1 µM CX-5461 imaged with a 20× objective. (**C**) HeyA8 cells were treated with either vehicle or CX-5461 (1 µM) for 6 h. Cells were stained with PicoGreen. Cells with cytoplasmic DNA were counted and quantified. (**D**). HeyA8 cells were treated the same as (**C**) and stained with anti-dsDNA, and anti-RNA:DNA (R-loop) antibodies. (**E**) COV362 cells were treated with vehicle, 1 µM CX-5461, 100 nM BMH-21, 5 nM paclitaxel, 50 µM carboplatin, and 250 µM temozolomide for 6 h and labeled with Sir-Hoechst. Quantitation of cytoplasmic positive cells is shown. The representative images are shown in Appendix A (* *p* = 0.0094, ** *p* < 0.0001). DNA build-up upon CX-5461 treatment, we labeled with Sir-Hoechst first to allow nuclear dye accumulation; we then washed away excess dye to obtain a “pre-labeled” nucleus. This would allow us to assess changes after CX-5461 treatment. COV362 cells were labeled with Sir-Hoechst for 2 h, visually verified for nuclear positivity, washed, and then media was added with CX-5461 (no additional Sir-Hoechst). After ~1 h of treatment, cells began displaying discrete, microsomal appearing cytoplasmic granules of Sir-Hoechst labeled DNA in the periphery of the nucleus (Figure 1B). Cells were imaged for 17 h at 15 min intervals (Appendix A for control and Appendix A for CX-5461-treated). To confirm the results in additional ovarian cancer cell lines, HeyA8 cells were treated with vehicle or CX-5461 for 12 h and stained with nucleic acid dye picogreen. Treatment of CX-5461 in HeyA8 cells also induced accumulation of cytosolic DNA (Figure 1C). Staining using anti-dsDNA antibody, and anti-RNA: DNA hybrid (R-loop) antibody further elucidate the nature of cytosolic buildup. Qualitatively, R-loop staining in the cytosol increased after treatment which is consistent with the recent publication showing CX-5461 inducing R-loop formation (Figure 1D).

**Figure 2 cancers-13-05056-f002:**
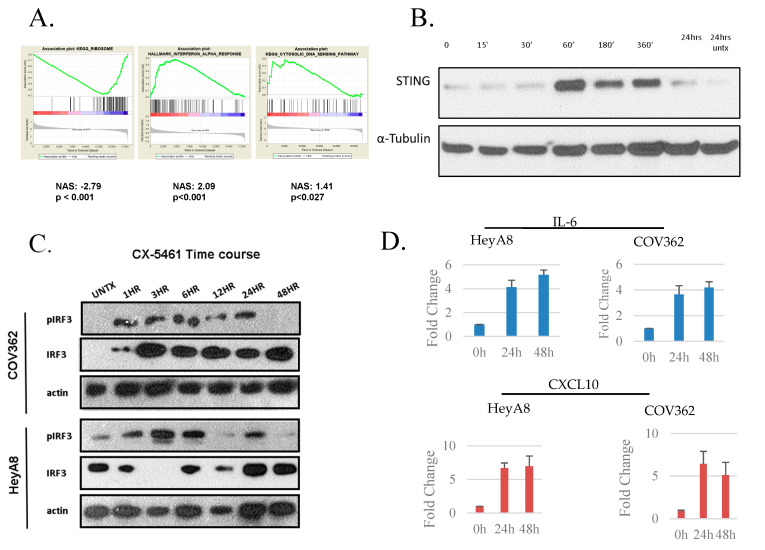
CX-5461-mediated cytosolic DNA induces the cGAS/STING system. (**A**) GSAAseqSP analysis of RNA-seq from 1 µM CX-5461 treated HeyA8 cells. (**B**) HeyA8 cells were treated with CX-5461 (1 µM) and cells were harvested at indicated time point and immunoblotted for STING. (**C**) COV362 and HeyA8 cells were treated with CX-5461 (1 µM) and cells were harvested at indicated time point and extracts were immunoblotted for STING for pIRF3 (S396), IRF3, and actin as internal control (The original Western blots have been shown in Appendix A). (**D**) COV362 and HeyA8 cells were treated with CX-5461 for 24 or 48 h, Cells were harvested and RNA were isolated using TRIZOL. RT-PCR of IL-6 and CXCL10 were performed. Normalized expression expressed as mean of three independent experiments ±SD.

**Figure 3 cancers-13-05056-f003:**
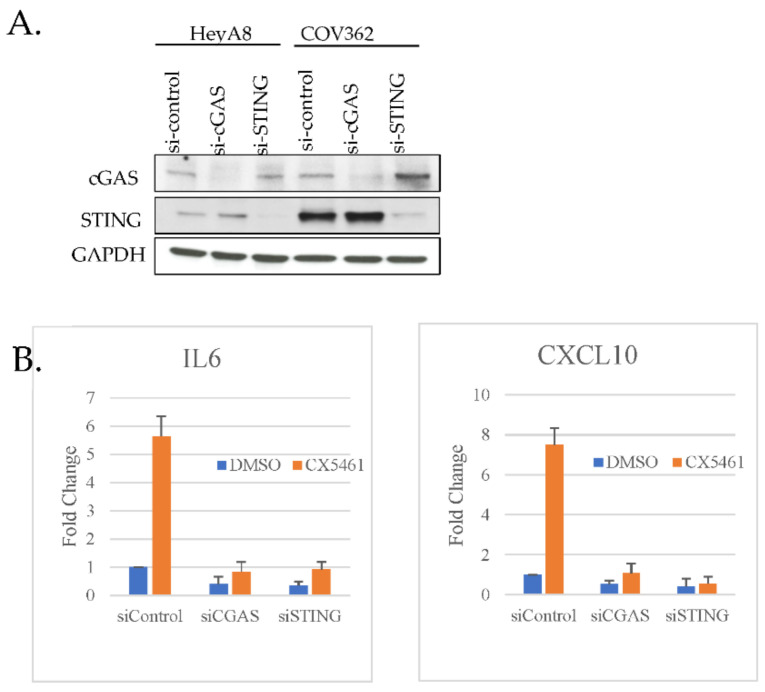
Knockdown of cGAS and STING abolishes CX-5461-mediated induction of IL-6 and CXCL10. (**A**) HeyA8 and COV362 cells were knocked down for either cGAS or STINGL with specific siRNA for 48 h. Cells were harvested and extracts were immunoblotted for cGAS, STING and GAPDH. (**B**) cGAS and STING were depleted in COV362 and HeyA8 cells as in (**A**) and RT-PCR of Il-6 and CXCL10 were performed.

**Figure 4 cancers-13-05056-f004:**
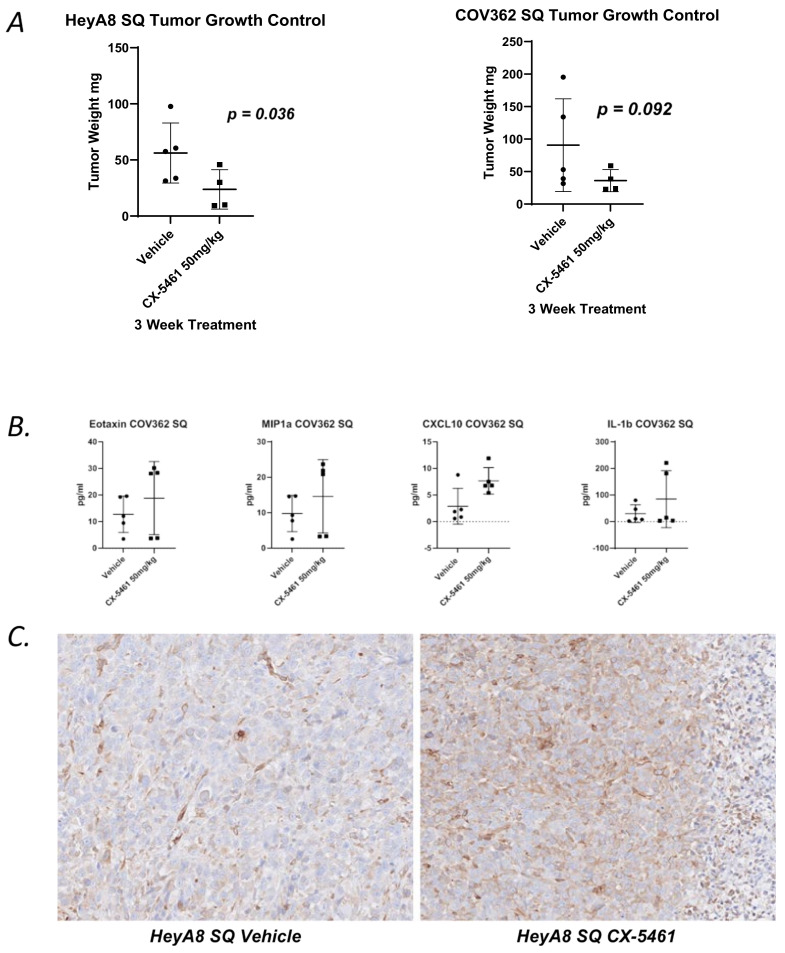
In vivo CX-5461-mediated cytosolic DNA induces the cGAS/STING system. (**A**) Percent change in tumor volume in HeyA8 SQ and COV362 SQ xenografts after 3 weeks of treatment with 50 mg/kg CX-5461 in groups of 5 nude mice. (**B**) Luminex of selected cytokines Eotaxin, MIP1a, CXCL10, and IL-1b after 3 weeks of treatment with 50 mg/kg CX-5461 in COV362 SQ and xenografts. (**C**) Immunohistochemistry of total STING in HeyA8 of SQ tumors after 3 weeks of treatment with 50 mg/kg CX-5461 or vehicle control at 20× magnification.

**Figure 5 cancers-13-05056-f005:**
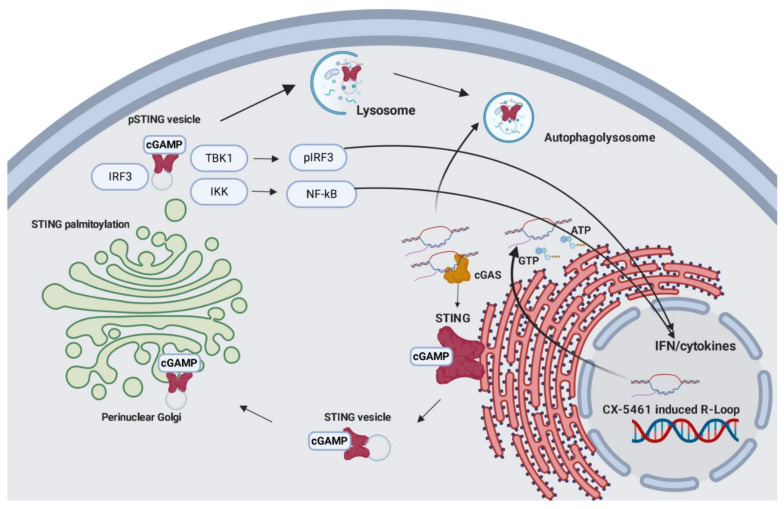
Graphical presentation of CX-5461-mediated activation of the cytosolic DNA sensing pathway. cGAS binding to the cytosolic DNA activates the secondary messenger 2,3-cGAMP using cGAS as a catalyst. Activated STING buds off the ER moving to the perinuclear Golgi, where it is palmitoylated. STING brings together TBK1, IRF3, and IKK. TBK1 phosphorylates STING, which leads to phosphorylation of IRF3 and activation of NFκB signaling through RelA. Both pathways lead to type I interferon activation, senescence-associated secretory phenotype (SASP), and interferon-stimulated gene (ISG) transcription.

## Data Availability

The data presented in this study are available in this article (and Appendix A).

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
