# Peer review of "CX-5461 Treatment Leads to Cytosolic DNA-Mediated STING Activation in Ovarian Cancer"

_cancers, 2021, doi:10.3390/cancers13205056_

Round 1

Reviewer 1 Report

Reviewing the manuscript entitled “CX-5461 Treatment Leads to Cytosolic DNA-Mediated STING 2 Activation in Ovarian Cancer”.  The authors evaluate the RNA polymerase I inhibitor CX-5461 effects on ovarian cancer cells. The results show that CX-5461 induces the release of cytoplasmic DNA that stimulates cGAS-STING signal- 12 ing, leading to the production of IFN type I in both cancer cells and xenografts in vivo. The described results and discussion are presented in a clear prototypical research approaches using an elegant experimental design. The introduction gives a narrative literature review and describes the focus of the study. The figures are very informative with very good quality.  The author does not oversell the finding and proposed further experiments to validate the presented results.  Although the finding needs further validation, the current manuscript has the potential for publication after minor revision.

  • I appreciate the fact that the authors address the limitation of the current study “ There are critical questions remain requiring further study. HeyA8 cells are wild type 262 for TP53, while the COV362 cells have the TP53 (Y220C) mutation. This could explain the 263 differences in responses since TP53 has been implicated in regulating immune responses, 264 as well as response to cytosolic DNA specifically. Further, work needs to be done to elu- 265 cidate the role of TP53 in response to CX-5461 and how the different mutations change the 266 outcome. The inflammatory phenotype may be a STING dependent activation, but it also 267 could be partly associated with a senescence associated secretory phenotype (SASP) as 268 CX-5461 induces senescence in many cell types. Most importantly, does the cancer speci- 269 ficity attributed to CX-5461 apply to this STING activation? As we saw variability in re- 270 sponse in different cell lines, a comprehensive analysis of multiple cell subtype and a bi- 271 omarker for response are needed”. 

  • In the discussion part, the author stated that “Our preliminary data have shown that the POL I inhibitor CX-5461 induces a signif- 234 icant accumulation of cytosolic DNA, transcriptionally activates STING, induces phos- 235 phorylation of IRF3, which induces type I IFN in ovarian cancer cells. Furthermore, CX- 236 5461 treatment decreases tumor burden in xenograft model. Our data uncovered that 237 DNA damage induced by CX-5461 generates cytosolic DNA, primarily dsDNA. CX-5461 238 treatment induces secretion of the type I interferon associated cytokines: IL-6 and 239 CXCL10 ”. This section needs to be related to reported literature and supporting with relevant references.

  • It would be a good idea to the chemical structure of CX-5461

Author Response

Response Letter

Dear Editors,

We would like to thank the reviewers for their thorough review and overall positive comments. We have addressed all of their suggestions as delineated below, and appreciate that it makes the manuscript stronger. Thank you for your continued consideration.

Sincerely,

Kuntal Biswas et. al.

Reviewer 1:

Reviewing the manuscript entitled “CX-5461 Treatment Leads to Cytosolic DNA-Mediated STING 2 Activation in Ovarian Cancer”.  The authors evaluate the RNA polymerase I inhibitor CX-5461 effects on ovarian cancer cells. The results show that CX-5461 induces the release of cytoplasmic DNA that stimulates cGAS-STING signaling, leading to the production of IFN type I in both cancer cells and xenografts in vivo. The described results and discussion are presented in a clear prototypical research approaches using an elegant experimental design. The introduction gives a narrative literature review and describes the focus of the study. The figures are very informative with very good quality.  The author does not oversell the finding and proposed further experiments to validate the presented results.  Although the finding needs further validation, the current manuscript has the potential for publication after minor revision.

  • I appreciate the fact that the authors address the limitation of the current study “There are critical questions remain requiring further study. HeyA8 cells are wild type for TP53, while the COV362 cells have the TP53 (Y220C) mutation. This could explain the differences in responses since TP53 has been implicated in regulating immune responses, as well as response to cytosolic DNA specifically. Further, work needs to be done to elucidate the role of TP53 in response to CX-5461 and how the different mutations change the outcome. The inflammatory phenotype may be a STING dependent activation, but it also 267 could be partly associated with a senescence associated secretory phenotype (SASP) as CX-5461 induces senescence in many cell types. Most importantly, does the cancer specificity attributed to CX-5461 apply to this STING activation? As we saw variability in response in different cell lines, a comprehensive analysis of multiple cell subtype and a biomarker for response are needed”. 
  • In the discussion part, the author stated that “Our preliminary data have shown that the POL I inhibitor CX-5461 induces a significant accumulation of cytosolic DNA, transcriptionally activates STING, induces phosphorylation of IRF3, which induces type I IFN in ovarian cancer cells. Furthermore, CX- 5461 treatment decreases tumor burden in xenograft model. Our data uncovered that DNA damage induced by CX-5461 generates cytosolic DNA, primarily dsDNA. CX-5461 treatment induces secretion of the type I interferon associated cytokines: IL-6 and CXCL10. This section needs to be related to reported literature and supporting with relevant references.

Response: Related reported literature was added in the discussion and relevant reference was inserted.

  • It would be a good idea to the chemical structure of CX-5461

Response: Link of the chemical structure of CX-5461 was added in the methods and materials section.

Reviewer 2 Report

The manuscript described the effect of an RNA polymerase inhibitor in trials for hematological cancers leads to cytosolic DNA-mediated STING activation in ovarian cancer. The results are interesting and may have promise in ovarian cancer treatment. However, authors used T test in the statistical analysis which is used for comparison of means between only two groups. Since there are more than two groups are involved in the experiments, ANOVA should be used instead of T-test. Also, in the materials section, the description of which cell lines are used, how were they cultured, and where were they obtained should be added.

Author Response

Response Letter

Dear Editors,

We would like to thank the reviewers for their thorough review and overall positive comments. We have addressed all of their suggestions as delineated below, and appreciate that it makes the manuscript stronger. Thank you for your continued consideration.

Sincerely,

Kuntal Biswas et. al.

Reviewer2:

The manuscript described the effect of an RNA polymerase inhibitor in trials for hematological cancers leads to cytosolic DNA-mediated STING activation in ovarian cancer. The results are interesting and may have promise in ovarian cancer treatment. However, authors used T test in the statistical analysis which is used for comparison of means between only two groups. Since there are more than two groups are involved in the experiments, ANOVA should be used instead of T-test. Also, in the materials section, the description of which cell lines are used, how were they cultured, and where were they obtained should be added.

Response: Thank you for your suggestion. We have changed our analysis to ANOVA from t-test. Rewrite the methods and exchanged the figure 1E.

Reviewer 3 Report

This manuscript demonstrates a previously unknown effect of CX-5461 treatment, shows for the first time that CX-5461 treatment may induce release of cytoplasmic DNA and stimulate STING signaling as well as some associated transcription programs in ovarian cancer cells. The results of this work have important clinical implications and are significant in unveiling a potentially powerful mechanism to exploit for increasing the efficacy of immunotherapies and leading to durable cancer cures. The manuscript is well written, the research is appropriately designed, and the results presented are convincing with reasonable and careful analysis. The work is recommended for publication as it is.

Author Response

Response Letter

Dear Editors,

We would like to thank the reviewers for their thorough review and overall positive comments. We have addressed all of their suggestions as delineated below, and appreciate that it makes the manuscript stronger. Thank you for your continued consideration.

Sincerely,

Kuntal Biswas et. al.

Reviewer 3:

This manuscript demonstrates a previously unknown effect of CX-5461 treatment, shows for the first time that CX-5461 treatment may induce release of cytoplasmic DNA and stimulate STING signaling as well as some associated transcription programs in ovarian cancer cells. The results of this work have important clinical implications and are significant in unveiling a potentially powerful mechanism to exploit for increasing the efficacy of immunotherapies and leading to durable cancer cures. The manuscript is well written, the research is appropriately designed, and the results presented are convincing with reasonable and careful analysis. The work is recommended for publication as it is.

Response: Thanks for your comments.

Reviewer 4 Report

Cornelison and collegues shows an interesting data regarding the CX-5461 downstream signaling modulating the transcription of cytokines.

Minor considerations

  • The author needs check carefully all the text and figures.

 Figure 1- The author should format all size, type and resolution of the font.  Moreover in fig 1C is missing some letters. Same observation for fig 2. Yet, some pictures are too small.

  • Pag 3 line 94- The author should add the informations of picogreen kit and Dapi used.

  • Pag 4 lines 137 and 139- the author should rename for C and C’or C1 and C2 (in the figure and legend).

  • I suggest the author indicate the number of experiments performed and number of images/cells analyzed.

  • Line159- The author should add de final dot after bibliography number. Check for all document.

  • Line 157- The author should patronize as Figure X or Fig X.

  • Line 187- The author should discuss the possibilities for STING activation started at 60’and ends with 360’.

  • Line 251- At the Discussion topic, the author hypothesize around types of cell death, what was not approached in this manuscript, but did’t mentioned the role of the cytokines analyzed.

  • In that sense, I suggest the author discuss how the cytokines analyzed could be involved with the favor outcome/ immune modulation.

  • Moreover, since it was cited at the simple summary (line 15), the author should discuss how this approach (CX-5461) could be used in combination with checkpoint inhibition.

Author Response

Response Letter

Dear Editors,

We would like to thank the reviewers for their thorough review and overall positive comments. We have addressed all of their suggestions as delineated below, and appreciate that it makes the manuscript stronger. Thank you for your continued consideration.

Sincerely,

Kuntal Biswas et. al.

Reviewer 4 Comments

Cornelison and collegues shows an interesting data regarding the CX-5461 downstream signaling modulating the transcription of cytokines.

Minor considerations

1) The author needs check carefully all the text and figures. Figure 1- The author should format all size, type and resolution of the font. Moreover in fig 1C is missing some letters. Same observation for fig 2. Yet, some pictures are too small.

Response: The issue was addressed in the current manuscript.

2) Page 3 line 94- The author should add the informations of picogreen kit and Dapi used.

Response: Picogreen and DAPI information were included in the method section.

3) Pag 4 lines 137 and 139- the author should rename for C and C’or C1 and C2 (in the figure and legend).

Response: Picogreen and DAPI are imaged in the same field. So, we named the both images as C.

4) I suggest the author indicate the number of experiments performed and number of images/cells analyzed.

Response: We have stated the number of experiments and how the images were analyzed in the methods and figure legends.

5) Line159- The author should add de final dot after bibliography number. Check for all document.

Response: We have changed.

6) Line 157- The author should patronize as Figure X or Fig X.

Response: We have changes all as Figure X.

7) Line 187- The author should discuss the possibilities for STING activation started at 60’and ends with 360’.

Response:  Added discussion section on the short time frame and its implication for synergistic therapy.

8) Line 251- At the Discussion topic, the author hypothesize around types of cell death, what was not approached in this manuscript, but didn’t mentioned the role of the cytokines analyzed.

Response: addressed in discussion specifically on STING activated IRF3 dependent mitotic cell death and inflammasome mediated cell death.

9) In that sense, I suggest the author discuss how the cytokines analyzed could be involved with the favor outcome/ immune modulation.

Response: addressed in discussion

10) Moreover, since it was cited at the simple summary (line 15), the author should discuss how this approach (CX-5461) could be used in combination with checkpoint inhibition.

Response: addressed in discussion